# Chronic Regulatory Focus and Work-Family Conflict among Chinese Workers

**DOI:** 10.3390/ijerph17124526

**Published:** 2020-06-23

**Authors:** Xinyuan (Roy) Zhao, Karthik Namasivayam, Nicholas J. Beutell, Jingyan Liu, Fujin Wang

**Affiliations:** 1Business School, Sun Yat-sen University, Guangzhou 510275, China; zhaoxy22@mail.sysu.edu.cn (X.Z.); ljy6633@126.com (J.L.); 2Saunders College of Business, Rochester Institute of Technology, Rochester, NY 14623, USA; karthiknamas@yahoo.com; 3LaPenta School of Business, Iona College, New Rochelle, NY 10801, USA; NBeutell@iona.edu

**Keywords:** chronic regulatory focus, work and family, challenge/hindrance strain

## Abstract

Survey data from 226 service employees were used to test the hypothesized moderating role of chronic self-regulatory focus on the relationships between work–family conflict (WFC) and challenge/hindrance strain. A follow-up scenario-based experiment (N = 93 executives) confirmed the results of the hypothesized model. Results from the two studies together demonstrated the moderating role of self-regulatory processes: chronic promotion-focused individuals perceived WFC as a challenge-type strain, while chronic prevention-focused individuals viewed WFC as a hindrance-type strain. Individuals use self-regulation strategically: in work domains, they regulate themselves so that family does not interfere with work. Individuals’ stress perceptions differ depending on the two dimensions of WFC as they regard interferences from (WIF) as a personal challenge, perhaps affording them an opportunity to balance work and life and to refine their abilities, but interferences from family to work (FIW) act as a barrier preventing them from achieving career success. When two-way interactions between WIF/FIW and chronic promotion/prevention foci were taken into consideration, the WIF/FIW main effects on challenge/hindrance stress became insignificant, suggesting that chronic self-regulation fully moderated the relationship. The results extend the current work–family research by incorporating self-regulatory processes as an important moderating variable, suggesting new research directions. The findings can help human resource management establish policies and benefit programs that take individual differences into account.

## 1. Introduction

The physiological model of personality traits has always been an important research field. According to Gray [1,2,3,4], personality can be divided into anxiety tendency and impulse tendency on the basis of different nervous systems and functions of the brain. Gray believes that different personality traits lead to diverse motivations and behaviors. Personality traits with an anxious tendency can lead to aversion motivation, while personality traits with an impulsive tendency can lead to desire motivation. On previous theoretical studies, Higgins proposed the regulatory focus theory stressors [5]. The regulatory focuses on an individual’s trait, which mainly reflects the characteristics to seek advantages and avoid disadvantages. The differences in such traits directly determine the distinction in individual cognition and behavior patterns [6]. Some studies about the effects of leadership on the psychological and behavioral effects have found that differences in employee traits directly influence how employees respond to leadership styles or behaviors [7]. As family-related issues become increasingly critical at the workplace [8,9], work–family conflict (negative spillover between work and family roles; WFC) often increases counter-productive attitudes and withdrawal behaviors among employees including emotional exhaustion and stress [10]. Work and family are two important areas of an individual’s social life [11]. When it is impossible to effectively balance the roles of family and work due to resource constraints (such as time and energy), there will be conflicts between the two roles [12]. Work family conflict is one of the important sources of stress [13] and it will decrease job satisfaction [14] and organizational citizenship behavior (OCB) [15], bring about job insecurity [16], role overload [17], and pressure [18], even leading to deviation from the organization and counterproductive behavior [19], thus affecting the individual’s job performance [20]. Previous studies have found that work–family conflict not only has a direct impact on job performance [21], but also indirectly affects job performance through mediating variables such as emotional exhaustion [22]. Extant findings about the relationship between WFC and stress levels, however, are not consistent [23]. Some scholars attribute these inconclusive study results to the omission of individual-level differences [24,25,26,27] and to samples that are not culturally diverse [28]. The current study responds to these concerns in four ways.

First, we identified chronic self-regulation as an important individual characteristic moderating WFC related stress perceptions. Individuals differ in their chronic regulatory focus and correspondingly, their means of dealing with external stressors [5,29]. Shi [30] proved that a situational regulatory focus had more extensive effects than a chronic regulatory focus on asymmetric perceptions of outcomes. Individuals with chronic promotion-focus tend to adopt positive patterns of behavioral and cognitive strategies; those with a chronic prevention-focus use conservative and preventive strategies. These two dimensions of regulatory focus may moderate the individual’s views about work–family interface to be seen as enrichment or conflict [31]. We extended this perspective to suggest that such differences causes individuals to adopt very different strategies toward perceiving and managing WFC, rather than solely positive attitudes (e.g., job satisfaction in [32]). Chronic promotion-focused individuals will perceive WFC as a challenge stress (as opportunities to balance work and family demands and achieving career and life success). On the other hand, individuals with chronic prevention-focused self-regulation experienced WFC as a hindrance stress (preventing them from fulfilling work and family responsibilities).

Second, we adopted and built on more recent research that suggests that individuals simultaneously perceive both challenge and hindrance stress with consequent effects on performance. When employees are forced to deviate from the normal or desired lifestyle, the uncomfortable feeling they experience and show is described as work pressure [33]. Employees who are under work pressure for a long time are more likely to be dissatisfied with their work and have worse work performance [34,35]. Too much work pressure can cause emotional exhaustion [36,37]. Work pressure may threaten the sustainable employability of the staff [38]. When direct leaders adopt ethical leadership, pay attention to the interests of employees, and provide support for leaders, employee anxiety can be effectively alleviated and they will behave better in their work [39,40]. Previous research has provided a two-dimensional conceptualization of the individual’s stress: workplace stressors are perceived as either a challenge (i.e., they view stressors as challenges or opportunities for personal achievement) or hindrance-related stress (i.e., they view stressors as obstacles or threats) [41,42,43,44]. The two types of stressors exist at the same time and affect the individual’s work performance. When one has more challenging stressors, he/she is more likely to achieve a satisfactory job performance [45]. Researchers have suggested that “combining and integrating challenge and hindrance appraisals would enable a more valid testing and a better understanding of the effect of a stressor on performance” [46]. We accordingly argue that individuals with different chronic self-regulatory modes will perceive challenge and hindrance types of WFC stress differently.

Third, responding to the complaints of inconsistent cultural effects and lack of research outside Western countries, this study collected data in China, a rapidly modernizing economy and society [47,48]. Researchers note that because of social networks (Guangxi) and Confucianism, hospitality employees in China, Korea, and other Asian countries experience work–family conflict and stress perceptions differently than their Western counterparts [49,50]. Other research has shown that the impact of job characteristics on work–family conflict was consistent and transportable across 48 countries [51]. Drawing on a Chinese hotel employee sample, this study contributes to an understanding of the impact of culture on WFC and stress perceptions [52,53].

A mixed method approach was adopted to ensure greater validity of the results and overcome the limitations of using either only a survey or experimentation to collect data [54,55]. The next section provides theoretical foundations and develops our hypotheses.

## 2. Theoretical Framework

### 2.1. Work–Family Conflict as a Salient Stressor

Work–family conflict (WFC) refers to incompatible interferences between the individual’s work and family roles [56] including two dynamic and reciprocal relationships with distinct antecedents and outcomes: work interfering with family (WIF) and family interfering with work (FIW) [57]. WIF and FIW have been identified as critical antecedents of employee stress levels relating to an individual’s experience of threat and unpleasantness [58,59]. Conflicts between work and family role demands relate to higher levels of stress [60]. Empirical research reports that WFC and general psychological stress (r = 0.29) [61] and WFC and physical stress (r = 0.26) were significantly associated [62]. It has also been reported that each dimension of WFC (WIF and FIW) correlated with work stress [63], and a meta-analytic review found the individual’s overall stress was positively associated with both WIF (r = 0.48) and FIW (r = 0.29) [64].

The theories about the relationship between work and family have experienced great development. For instance, boundary theory [65] points out that there is a boundary between work and life roles. Resource conservation theory [66] proposes that individuals tend to acquire and preserve resources by substituting and transferring resources. Based on the above theories, some scholars have proposed that role stress is an important factor leading to conflicts between work and family [67]. When employees undertake too much task load on a certain role, they will feel task-type pressure and tend to transfer existing resources between different roles [17]. Existing studies have not only verified the influence of antecedent variables (like role stress) on work–family conflict, but also explored the influence of work–family conflict on outcome variables such as work satisfaction, life satisfaction, and psychological depression [68].

Although research generally supports the impact of WFC on work outcomes, empirical studies have revealed mixed results. For example, Frone and colleagues [69] found an insignificant relationship between WFC and the individual’s health status. A meta-analysis reported a modest negative association of WFC with employee self-rated performance, but not supervisor-rated performance [46]. Some researchers have attributed the mixed results to inattention to individual characteristics that may moderate the relationship between WFC and work stress [61]. Gilboa et al. [46] hinted at the role of self-regulation, suggesting that the inconsistent findings probably revealed that individuals may engage in self-regulation, which influences their perceptions of WFC. 

Research has noted that different appraisals of WFC can lead to specific attitudinal and behavioral responses to stress. Cavanaugh et al. [70] suggest that some view stress as manageable, allowing for personal growth, but others perceive stress as unmanageable and an obstacle to growth. Thus, they argued that not all stress is negatively appraised by individuals. Wallace and colleagues [71] emphasize that stress evaluation varies by the individual’s means of appraising stressors. We extend these arguments and suggest that some individuals consider WFC as challenge-type stress that enhances work–family balance or enrichment, leading to positive work outcomes (e.g., better performance). Others may view WFC as a hindrance-type stress that creates obstacles to personal well-being and leads to negative work consequences (e.g., lower performance) [41,72,73]. In this paper, we identified the individual’s chronic self-regulatory focus as a critical moderator determining the likelihood that they perceive WFC as challenge or hindrance stress.

### 2.2. The Moderating Role of Chronic Promotion/Prevention-Focus

Individuals adopt promotion or prevention self-regulation strategies relatively, which consistently [5] influence work attitudes such as organizational commitment [74]. Individuals with chronic promotion-focus concentrate on achieving desired end-states, while those with a chronic prevention-focus avoid undesired end-states [75]. Accordingly, chronic promotion-focus may lead individuals to perceive WFC as challenge-type stress, whereas chronic prevention-focus may lead individuals to view WFC as hindrance-type stress.

#### 2.2.1. Accessibility

When facing conflicts, promotion-focused people are more likely to access attitudes that relate to successful accomplishment of work tasks and family demands and view WFC as a challenge stressor [75]. Conversely, chronically prevention-focused persons are more likely to access attitudes that support the management and reduction of unwelcome role conflicts between work and family. These individuals consider conflicts as hindrances and obstacles to the fulfillment of personal goals.

#### 2.2.2. Selective Resource Allocation

The ‘scarcity perspective’ assumes that individuals have finite psychological (cognitive) and physiological (e.g., time and energy) resources and employ self-regulatory processes to allocate such finite resources to resolve WFC [56,76,77]. Promotion-focused individuals consider conflicts as challenge-type stress and this allows them to self-consciously allocate resources in a manner aimed at balancing and mutually facilitating work and family domains. In contrast, individuals with a chronic prevention-focus consider conflicts as hindrances and adopt a conservation strategy to passively (reactively) allocate resources aimed at conserving finite resources and eliminating, rather than resolving, hindrances.

## 3. Hypothesis

Taken together, we propose that chronic regulatory focus will influence the employee’s perception of WFC (both WIF and FIW) as either a challenge or hindrance stress (see Figure 1). Accordingly,

**Hypothesis** **1a.**
*Employees with chronic promotion-focus will perceive WIF as a challenge, but not hindrance stress.*


**Hypothesis** **1b.**
*Employees with chronic promotion-focus will perceive FIW as a challenge, but not hindrance stress.*


**Hypothesis** **2a.**
*Employees with chronic prevention-focus will perceive WIF as a hindrance, but not challenge stress.*


**Hypothesis** **2b.**
*Employees with chronic prevention-focus will perceive FIW as a hindrance, but not challenge stress.*


Chronic regulatory focus, considered a moderator in this paper, is distinct from coping strategies adopted in previous WFC studies. First, coping is considered a moderator in previous work–family research [78], although it reflects the individual’s cognitive and behavioral responses (consequences) to WFC [79]. In contrast, the individual’s chronic regulatory focus is more relevant to self-guidance in their reactions to WFC [5,80]. Second, coping is situation-specific, whereas chronic regulatory foci are a general guide to the individual’s behavioral responses. Third, individuals facing stress may cope in one of two ways: adopting active (or problem-focused) or avoidance (or emotion-focused) strategies [81]. Self-regulation, on the other hand, motivates a more complete response: individuals with a promotion-focus attempt to employ their full potential to achieve success in both work and family domains, while prevention-focused individuals attempt to avoid failures in either domain [5,80].

## 4. Method

Responding to calls for more triangulation and multiple samples in research projects, we address the hypotheses in two studies: (1) a field survey, and (2) a scenario-based experiment. In Study 1, we collected the hotel employees’ responses to the study variables using a survey, and in Study 2, the managers responded to two written scenario statements depicting high or low work–family conflict situations.

### 4.1. Study 1

Participants were recruited from eight hotels. Long work hours, irregular and inflexible work schedules, heavy workloads, and low wages characterize the hospitality industry, leading to WFC and consequent alcohol abuse and higher divorce rates [82]. Therefore, hotel jobs are an appropriate occupation for investigating WFC issues [28,83]. The current study investigated employees at all levels: front-line employees and managers. The sample included married and single individuals, with or without children living at home, and with or without an employed spouse because all employees have the potential to experience work–family conflicts [83].

Researchers note that only 19.9% of the WFC studies have been conducted outside of the U.S. [28]. The present study contributes to the WFC literature by testing it in a Chinese context. About 80% of urban Chinese women between the ages of 16 and 54 are employed, accounting for nearly 40% of the total urban labor force. Over 90% of urban families are headed by working parents, indicating that China may have similar dual-earner family issues as the USA. [84]. The nature and structure of work and family demand is not significantly different in China compared to the USA [85], ensuring that it is an appropriate setting for WFC research.

In each hotel, forty employees were randomly selected from a list provided by the human resource department and three hundred and twenty questionnaires were distributed. Employees were informed that the research was for academic purposes and their participation was voluntary and confidential. They were allowed a week to fill out and return completed surveys in individually sealed envelopes to a collection box in their respective human resource departments. Validity and reliability was ensured by back-translating the measures [85].

Valid surveys were returned by 226 employees from eight hospitality organizations (response rate = 70.6%; surveys with missing values were excluded; see Table 1). A total of 57.1% of the subjects were female, most respondents (89.8%) were younger than 40 years of age (M = 28.72, SD = 7.90), and the average number of dependents (child, parents, relatives) was 1.14 (SD = 1.70). Most of the respondents had a high school (29.6%) or a college education (35.4%). A total of 55.8% of the sample was single and 37.2% were married. They had on average 6.82 years of work experience (SD = 7.28) and 40.3% were frontline employees and 41.6% were managers.

### 4.2. Study 2

Ninety-three hospitality managers were recruited from several business-related seminars (e.g., leadership workshops) in China (response rate = 93%; see Table 2). Participants were mainly males (75.3%), on average 32 years old (SD = 5.15), and married (71%). Most had a college (44.1%) or graduate education (35.5%), about ten years work experience (M = 9.60, SD = 5.43), and the mean number of dependents was 4.52 (SD = 1.10). To ensure comparability of the findings, the same measures were used in both studies.

Participants were provided with a written scenario [10] adapted to fit the study context. Participants imagined that they were the focal person in a business situation and were randomly assigned to two conditions: low or high work–family conflict. Specifically, some participants were assigned to a scenario involving a work–family conflict situation while others were assigned to a scenario depicting work–family balance (the manipulated information is in italics): “You are an employee at ABC hotel [originally ‘company’]. Recently, a number of important projects have landed on your boss’s desk. This morning, in a meeting with your boss, he assigned one of these projects to you. Although the project entails various responsibilities, in looking over the projects you are currently working on, you realize that this additional project will require you to spend much more time at work/will not require you to spend much more time at work. Therefore, you will have less time/about the same amount of time to spend on activities in your personal life that are very important to you” (10: 20).

Finally, participants provided responses to the study variables and manipulation checks. WFC manipulations were evaluated in two ways: [86] WFC perceptions were evaluated by the average value of WIF and FIW questions (i.e., “Your family burdens will prevent you from accomplishing the assigned work project” and “Accomplishing the assigned work project will prevent you from taking care of your family”); and (2) a measure of work–family balance perception (“In your mind, your family life and work tasks are balanced”). One item was also provided for the participants to rate scenario realism. Responses were provided on a seven-point Likert scale from 1 = strongly disagree to 7 = strongly agree.

## 5. Measures

As the studies were conducted in a new context, a series of exploratory factor analysis (EFA) were conducted to identify representative items (see Table 3).

### 5.1. Work–Family Conflict

Work–family conflict was measured with eleven items [87]. Six items measured work interfering with family (WIF; e.g., my job keeps me from spending time with my family members); five items measured family interfering with work (FIW; e.g., my family demands make it hard for me to do my job well). We selected six typical questions among those items (see Table 3). Responses were collected on a seven-point Likert scale anchored with 1 = strongly disagree and 7 = strongly agree.

### 5.2. Chronic Regulatory Focus

The general regulatory focus measure (GRFM; [88]) identified the individual’s preferred regulatory focus mode. The GRFM inventory had eighteen items (nine to measure chronic promotion focus and nine for chronic prevention focus). For example, a question measuring chronic promotion focus is “in general, I am focused on achieving positive outcomes in my life”; chronic prevention focus was measured by questions such as: “I frequently think about how I can prevent failures in my life.” We selected six typical questions among those items (see Table 3). Responses were collected on a seven-point Likert scale from 1 = strongly disagree to 7 = strongly agree.

### 5.3. Challenge/Hindrance Stress

The participant’s perceptions of challenge/hindrance stress were measured with eleven items adapted to the context of work–family issues [70]. Six questions measured the individual’s challenge stress (e.g., “dealing with a large amount of workload and family responsibilities can inspire my full potential”) and five items assessed the individual’s hindrance stress (e.g., “incompatible issues between my work and family prevent me from doing job well”). We selected five typical questions among those items (see Table 3). Responses to questions about challenge stress were measured on a scale anchored by 1 = least or very little and 7 = most or very much, and responses to questions about hindrance stress were measured on a scale anchored by 1 = impossible in my job and 7 = very possible in my job.

First, psychometric properties including convergent and discriminant validity of all multi-item constructs were evaluated using a confirmatory factor analysis. Hierarchical regression analyses (HRA) then tested the moderation effects of chronic promotion and prevention foci on the relationships between WIF/FIW and challenge/hindrance stress. Following Aiken and West [89], simple slope analyses (SSA) were conducted to examine the moderation role of chronic promotion or prevention-focus in more detail; compared to a traditional graphical approach, this procedure improves the power of the statistical tests [90].

## 6. Results

### 6.1. Descriptive Statistics and Manipulation Check

Table 4 presents the correlation matrix with means and standard deviations. Consistent with previous research [57,91,92], the employee’s average level of WIF (M = 4.15, SD = 1.42) was greater than that of FIW (M = 2.39, SD = 1.14), and WIF and FIW were correlated (r = 0.13, *p* = 0.05). WIF positively relates to both challenge (r = 0.16, *p* = 0.01) and hindrance stress (r = 0.11, *p* = 0.09), but FIW did not. At the same time, the bi-variate correlations support the proposed notion that chronic promotion and prevention foci have various patterns of moderation. Chronic promotion-focus relates negatively to FIW (r = −0.19, *p* < 0.01), but not WIF (r = −0.03, *p* > 0.10), while chronic prevention-focus relates positively to both FIW (r = 0.24, *p* < 0.01) and WIF (r = 0.16, *p* = 0.02). Chronic promotion-focus relates positively to challenge stress (r = 0.26, *p* < 0.01); chronic prevention focus, conversely, relates to hindrance stress (r = 0.14, *p* = 0.04).

The descriptive findings of Study 2 are reported in Table 1. They resemble those of Study 1, however, with greater coefficient values and a significant relationship between FIW and hindrance stress (r = 0.43, p < 0.01). Similar reported mean values for chronic promotion and prevention foci in both studies indicated that respondents in both groups exhibited similar deep-level characteristics and were homogeneous. One-way ANOVAs (IV: scenario type; DVs: WFC perceptions, work–family balance perceptions, and realism) were conducted to check manipulations (F = 9.81, *p* < 0.01; F = 5.88, *p* < 0.02). High WFC condition participants responded with higher conflict scores (M = 3.97, SD = 1.87) and lower balance scores (M = 4.53, SD = 1.60) compared to participants in low WFC context (conflict: M = 2.79, SD = 1.74 and balance: M = 5.30, SD = 1.46). The mean rating on the scenario realism was 4.86 (SD = 1.12). These results demonstrate that the scenarios effectively framed the participants’ levels of WFC.

### 6.2. Measurement Tests

Scale reliabilities (Cronbach’s α) of Study 1 and 2 (see Table 1) were adequate, ranging from 0.70 to 0.86, except for a slightly low, but acceptable value (0.65) of hindrance stress in Study 2 [93,94]. For all scales, the composite scale reliability (CR) values (ranging from 0.73 to 0.90) were above the cut-off of 0.70 and the average variance extracted (AVE) values (ranging from 0.55 to 0.76) exceeded the 0.50 cut-off [95]. As shown in Table 1, the partial correlations were lower than the square root of the AVE, demonstrating adequate discriminant validity [96]. The CFA results of the two six-factor measurement model confirmed that the measurements fit the sample adequately (Study 1: χ^2^
_(696)_ = 939.71, GFI = 0.92, AGFI = 0.89, CFI = 0.94, IFI = 0.94, RMSEA = 0.04; and Study 2: χ^2^
_(104)_ = 136.38, GFI = 0.93, AGFI = 0.90, CFI = 0.94, IFI = 0.94, RMSEA = 0.06).

### 6.3. Testing Moderation Effects

Table 5 presents the hierarchical regression analysis (HRA) results. At the first step, testing only the main effects, the influence of chronic promotion-focus (CPM) (Study 1: B = 0.23, *p* < 0.01; Study 2: B = 0.20, *p* < 0.01) and of WIF (Study 1: B = 0.12, *p* = 0.01; Study 2: B = 0.15, *p* = 0.02) on challenge stress were significant in both Study 1 and 2; no other main effects were found. Both FIW (Study 1: B = 0.08, p = 0.26; Study 2: B = 0.37, *p* < 0.01) and chronic prevention-focus (CPV) (Study 1: B = 0.07, *p* = 0.12; Study 2: B = 0.18, *p* = 0.06) influenced hindrance stress significantly only in the experiment, but not in the field survey, while the main effects of WIF and CPM were insignificant in both studies.

Next, the two-way interaction terms (i.e., WIF × CPM, WIF × CPV, FIW × CPM and FIW × CPV) were tested. To create the interaction terms, the variables were first centered to their means and then were multiplied with corresponding ones. FIW × CPV had a significant influence on challenge stress in both studies (Study 1: B = 0.10, *p* < 0.01; Study 2: B = 0.11, *p* = 0.04). Surprisingly, WIF × CPV had a significant impact on challenge stress in Study 1, but not in Study 2 (Study 1: B = −0.07, *p* < 0.01; Study 2: B = 0.05, *p* = 0.23). Other interaction terms were insignificant.

### 6.4. Follow-Up Examinations of Moderation Effects

As noted earlier, simple slope analyses were conducted (see Table 6). First, WIF related significantly to challenge stress for individuals with high and low levels of chronic promotion-focus. That is, for individuals with chronic promotion-focus, greater WIF leads to stronger challenge stress. Thus, Hypothesis 1a is supported.

Second, individuals with high, compared with low, chronic prevention-focus reported a greater negative relationship of FIW to challenge stress. That is, individuals with high chronic prevention focus are less likely to perceive FIW as hindrance stress rather than challenge stress. Thus, Hypothesis 2b is supported.

Third, individuals with low prevention-focus rather than those with high prevention-focus reported stronger positive relationship of WIF to challenge stress. That is, employees with low chronic prevention-focus are more likely to perceive WIF as challenge stress. However, this finding was not supported in Study 2. Finally, the insignificant results of HRA and SSA analyses indicate that chronic promotion- and prevention-focus have no moderating role when hindrance stress was the dependent variable. Therefore, Hypotheses 1b and 2a were not supported.

## 7. Discussion

Overall, Hypothesis 1a and 2b were supported but Hypotheses 1b and 2a were not. As stated, studies have found ambivalent results for the relationships between WFC and stress. We attempted to clarify the relationships by adopting a two-dimensional—challenge and hindrance—view of stress. The results of this study indicate that the individual’s stress perceptions differ depending on the two dimensions of WFC. Individuals in this study regarded interferences from work to family (WIF) as a personal challenge, perhaps affording them an opportunity to balance work and life and to refine their abilities, but interferences from family to work (FIW) as a barrier prevented them from achieving career success. These different relationships between stress type and the two dimensions of WFC may result from how individuals interpret WFC and its consequences. For example, increased work to family interference may require individuals to allocate more time and effort to work such as working overtime. As breadwinners, individuals need to hold on to a job to ensure quality of life and well-being. When an individual’s family role demands exceed their time, effort, and other personal resources, individuals may perceive higher levels of hindrance stress caused by the work role.

The study also revealed that when two-way interactions between WIF/FIW and chronic promotion/prevention foci were taken into consideration, the WIF/FIW main effects on challenge/hindrance stress became insignificant, suggesting that chronic self-regulation fully moderates the relationship. Individuals chronically prone to adopting promotion-focused self-regulation strategies as well as those with low levels of chronic prevention-focus, report WIF as challenge stress; individuals with high levels of chronic prevention focus are less likely to identify FIW as challenge stress.

The findings permitted us to unpack the individual’s stress evaluation processes and consequent attitudinal outcomes. When family roles and demands interfere with their work roles, the individual’s self-regulatory strategies come into operation. This may be because family roles are non-negotiable and individuals focus on work roles to interpret and appraise the conflict landscape. Self-regulation may be important in this case because it may be considered illegitimate by the organization to allow family roles to interfere with work roles [46] and individuals may need to find behavioral remedies. Individuals with high levels of chronic promotion-focus and low levels of prevention-focus may, accordingly, consider WIF a personal challenge, providing an opportunity to actively and creatively solve the conflicts. In contrast, individuals with high levels of prevention-focus may respond with apprehension to the family–work demands and negatively evaluate their jobs, since, as noted, individuals tend to ‘blame’ their jobs for any discomfort. We can speculate from these results that the individual’s performance and organizational outcomes depend on self-regulatory processes. More research detailing the cognitive pathways and strategies individuals adopt is required to fully account for the antecedents of stress, for example WFC, on organizational outcomes.

Some research has noted that there is little variance in patterns of work and family demands between the USA and China (e.g., [85]). In contrast to study results in Western contexts, participants perceived negative work–family spillover as challenge stress rather than hindrance stress. Chronic self-regulation may be considered a proactive psychological coping mechanism and it will be interesting to see if the elicitation of self-regulatory processes depends on the relative importance of work and family demands. For example, compared to the USA, China represents an Oriental culture with Confucianism as the basis of social values [84]. Chinese employees may worry that family role interference can influence their job performance, eliciting greater attention to self-regulatory processes to reduce conflicts. Future research can examine how culture influences self-regulation in a WFC context.

Responding to calls in the literature, WFC was examined in the hotel industry. High customer-contact services influence organizational mandates on employee emotional/behavioral displays. The effect of such organizationally mandated behavior is an interesting research pursuit as it connects to notions of self-regulation and emotional labor [97,98]. Such research may also help resolve whether individuals in certain industries are more prone to adopting distinct patterns in their challenge/hindrance stressor appraisals, irrespective of their self-regulatory focus.

Previous studies have seldom discussed the moderating role of chronic self-regulatory focus on the relationships between work–family conflict (WFC) and challenge/hindrance strain. Experiments in the study have found that chronic promotion-focused individuals perceive WFC as challenge-type strain, while chronic prevention-focused individuals view WFC as a hindrance-type strain.

## 8. Practical Implications

This study provides inspiration for organizations to help guide employees to coordinate WFC. Human resource management should establish policies and benefit programs that take individual differences into account. Family-friendly benefits are offered by organizations, but they are not always utilized by employees [99,100]. This study revealed that many individuals value their jobs and regard WFC as a challenge stress, using self-regulation to reduce interference between work and family [101]. This may be one reason why organizational benefits are under-utilized. To remedy this, employee assistance policies can provide assistance to individuals on the use of self-regulation strategies. For example, organizations can reward and recognize employees who successfully fulfill work and family responsibilities, widely disseminate such experiences in the organization, and establish those employees as behavioral role models. Organizations can provide time-management training to strengthen employee efficacy in successfully managing their work and life domains.

Organizations should consider the HRM system, work practices, and organizational culture to reduce WFC [61]. Organizations can provide more work resources (e.g., telework), family-related instrumental support (e.g., assistance with child-care responsibilities), and resources for individuals to fulfill family obligations (e.g., paid leave for taking their children to the hospital) [78]. These social supportive efforts may serve to increase the individual’s work engagement and job satisfaction.

## 9. Limitations

With respect to the experimental design, although realism and manipulation checks were conducted, it is possible that a discrepancy exists between the actual and experimental experiences of WFC and work stress. Second, the interference between work and family domains of hotel employees or the WFC perceptions of college student samples may be different than those of individuals in other industries. Finally, this study was conducted in China. Cultural differences may account for some of these results; further research in other cultural settings will help validate the findings of this study.

## 10. Conclusions

This study examined the effect of individual level differences (i.e., chronic promotion vs. prevention-focus) on the relationship between WFC and the individual’s perceptions of stress. We first extended existing WFC research by examining the moderation role of the individual’s self-regulatory focus. Our findings suggest that individuals use self-regulation strategically: in work domains, they regulate themselves so that family does not interfere with work. We then discussed a two-dimensional view (challenge/hindrance stress) of stress appraisal. Next, responding to the call to extend WFC research across different occupations and non-Western cultural backgrounds, this study was conducted in China and in the hospitality industry. Finally, a mixed method data collection strengthened the study.

## Figures and Tables

**Figure 1 ijerph-17-04526-f001:**
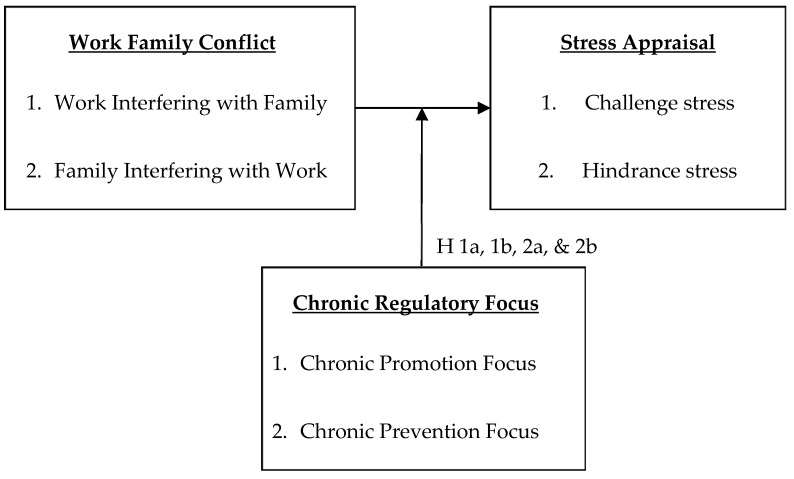
The conceptual model. H = Hypothesis.

**Table 1 ijerph-17-04526-t001:** Sample distribution of Study 1.

Items	Categories	Respondents	Percentage (%)
Management	Front-line employees	91	40.3
	Managers	94	41.6
	Other level	41	18.1
Countries	America	181	80.1
	Outside of America	45	19.9
Gender	Female	129	57.1
	Male	97	42.9
Age	Under 40	203	89.8
	Over 40	23	10.2
Educational background	High school	67	29.6
	College	80	35.4
	Other background	79	34.9
Marital status	Single	126	55.8
	Married	84	37.2
	Other status	16	7.1

**Table 2 ijerph-17-04526-t002:** Sample distribution of Study 2.

Items	Categories	Respondents	Percentage (%)
Gender	Female	23	24.7
	Male	70	75.3
Educational background	College	41	44.1
	Graduate education	33	35.5
	Other background	19	20.4
Marital status	Married	66	71
	Other status	27	29

**Table 3 ijerph-17-04526-t003:** Items and loading factors.

Constructs	Items	Study 1	Study 2
WIF	When I get home from my job, I do not have the energy to do work around the house	0.892	0.855
	I spend so much time working that I am unable to get much done at home	0.722	0.629
	After work, I am often too tired to do things with my spouse or partner	0.676	0.795
FIW	The demands of my family life limit the number of hours I’m able to work	0.856	0.801
	My family make it hard for me to do my job well	0.781	0.684
	I’m so tired from all the things I have to do at home that it’s hard to have the energy to do my job	0.729	0.778
CPM	I frequently imagine how I will achieve my hopes and aspirations.	1.12	0.721
	I often think about the person I would ideally like to be in the future.	0.457	0.949
	I typically focus on the success I hope to achieve in the future.	0.752	0.764
CPV	I often think about the person I am afraid I might become in the future.	0.726	0.769
	I am anxious that I will fall short of my responsibilities and obligations.	0.754	0.816
	In general, I am focused on preventing negative events in my life.	0.741	0.543
CST	The number of projects and/or assignments I have	0.888	0.768
	The amount of time I spend at work	0.829	0.792
	The volume of work that must be accomplished in the allotted time	0.721	0.794
HST	The lack of job security I have	0.947	0.717
	The degree to which my career seems “stalled”	0.591	0.785

Note. Study 1: N = 287, and Study 2: N = 93. WIF = work interfering with family; FIW = family interfering with work; CPM = chronic promotion focus; CPV = chronic prevention focus; CST = challenge stress; HST = hindrance stress.

**Table 4 ijerph-17-04526-t004:** Descriptive statistics.

	Study 1						Study 2											
Variable	M	SD	α	CR	AVE	Sq-AVE	M	SD	α	CR	AVE	Sq-AVE	1	2	3	4	5	6
1. WIF	4.15	1.42	0.81	0.82	0.60	0.77	3.80	1.78	0.86	0.90	0.76	0.87	—	0.42 **	0.26 *	0.20	0.07	0.23 *
2. FIW	2.39	1.14	0.82	0.82	0.61	0.78	2.57	1.39	0.78	0.86	0.67	0.82	0.13 *	—	0.11	0.43 **	−0.07	0.26 *
3. CST	5.01	1.02	0.83	0.82	0.65	0.82	4.95	1.05	0.83	0.90	0.74	0.86	0.16 *	−0.04	—	0.09	0.19	−0.08
4. HST	3.91	1.08	0.75	0.76	0.63	0.79	2.64	1.34	0.65	0.85	0.74	0.86	0.11 ^+^	0.10	0.03	—	−0.01	0.29 **
5. CPM	5.64	1.17	0.73	0.82	0.73	0.86	5.62	1.03	0.71	0.81	0.59	0.77	−0.03	−0.19 **	0.26 **	0.05	—	0.09
6. CPV	3.76	1.55	0.73	0.71	0.55	0.74	3.48	1.44	0.70	0.88	0.71	0.84	0.16 *	0.24 **	0.05	0.14 *	0.01	—

Note. Study 1: N = 287, and Study 2: N = 93. M: mean; SD: standard deviation; CR: composite reliability; AVE: average variance extracted; Sq-AVE: the square root of the average variance extracted; WIF = work interfering with family; FIW = family interfering with work; CPM = chronic promotion-focus; CPV = chronic prevention-focus; CST = challenge stress; HST = hindrance stress. Correlation coefficients of Study 1 (survey) are below diagonal, and those of Study 2 (manager experiment) are above diagonal. + *p* < 0.10; * *p* < 0.05; ** *p* < 0.01.

**Table 5 ijerph-17-04526-t005:** Regression analysis.

		Study 1				Study 2			
		DV = CST		DV = HST		DV = CST		DV = HST	
		Step 1	Step 2	Step 1	Step 2	Step 1	Step 2	Step 1	Step 2
		B(S.E.)	B(S.E.)	B(S.E.)	B(S.E.)	B(S.E.)	B(S.E.)	B(S.E.)	B(S.E.)
Main effects	WIF	0.12(0.05) *	0.07(0.23)	0.07(.05)	0.11(0.26)	0.15(.07) *	0.47(0.45)	−0.01(0.08)	−0.04(0.58)
	FIW	−0.02(0.06)	−0.12(0.29)	0.08(0.07)	0.25(0.32)	0.05(.09)	-0.10(.53)	0.37(0.10) **	0.48(0.69)
	CPM	0.23(0.06) **	0.10(0.19)	0.06(0.06)	0.14(0.22)	0.20(.10) *	0.64(0.23) **	<0.01(0.12)	0.14(0.30)
	CPV	0.02(0.04)	0.11(0.13)	0.07(0.05)	0.12(0.15)	−0.13(.08)	−0.65(0.19) **	0.18(0.09) *	−0.01(0.24)
Interaction terms	WIF × CPM	—	0.06(0.04)	—	0.00(0.04)	—	−0.08(0.08)	—	−0.01(0.10)
	WIF × CPV	—	−0.07(0.03) **	—	-0.02(0.03)	—	0.05(0.04)	—	0.03(0.06)
	FIW × CPM	—	−0.05(0.04)	—	−0.04(0.05)	—	−0.06(0.09)	—	−0.04(0.12)
	FIW × CPV	—	0.10(0.04) **	—	0.01(0.04)	—	0.11(0.05) *	—	0.02(0.07)
	Model F	5.87 **		2.03 ^+^	1.09	3.14 *	3.67 **	6.22 **	3.12 **
	Total R^2^	0.10	0.15	0.04	0.04	0.13	0.26	0.22	0.23
	Adjusted R^2^	0.08	0.12	0.02	0.00	0.09	0.19	0.19	0.16
	∆R^2^	—	0.05 **	—	0.00	—	0.13 **	—	0.01

Note: The abbreviations follow those in Table 4. + *p* < 0.10; * *p* < 0.05; ** *p* < 0.01.

**Table 6 ijerph-17-04526-t006:** Simple slope analysis.

			Study 1			Study 2	
Y	X	Moderator	Level	(X’s) B	p	(X’s) B	p
CST	WIF	CPM	High	−0.05	0.08 ^+^	0.88	0.03 ^*^
			Low	−0.14	0.60	0.65	0.02 ^*^
		CPV	High	0.26	<0.01 ^**^	−0.26	0.18
			Low	0.46	<0.01^**^	<0.01	0.99
	FIW	CPM	High	0.16	0.49	0.75	0.13
			Low	0.24	0.44	0.54	0.12
		CPV	High	−0.48	0.03 ^*^	-0.56	0.04 ^*^
			Low	−0.24	0.03 ^*^	−0.20	0.17
HST	WIF	CPM	High	0.13	0.50	0.59	0.27
			Low	0.16	0.60	0.46	0.22
		CPV	High	0.10	0.23	−0.06	0.83
			Low	0.15	0.38	0.04	0.74
	FIW	CPM	High	0.25	0.28	0.70	0.23
			Low	0.33	0.34	0.61	0.14
		CPV	High	0.06	0.61	0.17	0.61
			Low	0.05	0.83	0.27	0.11

Note. The abbreviations follow those in Table 4. The regression coefficients of predictors (Xs) with p-values are given in the table. + *p* < 0.10; * *p* < 0.05; ** *p* < 0.01.

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
