# Peer review of "Chronic Regulatory Focus and Work-Family Conflict among Chinese Workers"

_ijerph, 2020, doi:10.3390/ijerph17124526_

Round 1
Reviewer 1 Report
This is a good piece of paper. Thanks to the author for their efforts. I have gone through the whole paper. I have some questions regarding methodology and literature review.
First author must focus on the research gap mentioning goals relating to prior literature.
In the hypothesis development literature review is very poor therefore it is necessary to discuss related literature of each hypothesis.
Author should use a table for the discussion of data collection.
Variable discussion is very limited. It is necessary to describe each and every variable according to the question. (See reference).
Moderating variable issues ignore the necessity of it and the implications of the results.
Finally, you must use whether your hypothesis are accepted or rejected very clearly in the results also in the discussion part. Moreover, implications part is very generous therefore it is necessary to revised strongly in the relevant field.
Include the questionnaire in the appendix along with loading factors.
https://www.mdpi.com/1660-4601/16/22/4559
Author Response
Point 1: Author should use a table for the discussion of data collection.
Response 1: Two tables for sample distribution of study 1 and study 2 has been added in line 223 and line 253. Table 1 consists of management,countries,gender,age,educational background and marital status.Table 2 consists of gender,educational background and marital status.According statistics has been corrected in line 216, 217, 219, 220, 221, 227, 228, 229.
Point 2: You must use whether your hypothesis are accepted or rejected very clearly in the results also in the discussion part.
Response 2: Conclusion has been add in the discussion part as "Hypothesis 1a & 2b were supported. Hypotheses 1b & 2a were not supported."
Point 3: Variable discussion is very limited. It is necessary to describe each and every variable according to the question. Include the questionnaire in the appendix along with loading factors.
Response 3: Measurement items and loading factors have been displayed by table 3 in line 282.
Items for Work interfering with family (WIF) are "When I get home from my job, I do not have the energy to do work around the house" "I spend so much time working that I am unable to get much done at home" "After work, I am often too tired to do things with my spouse or partner".
Items for family interfering with work (FIW) are "The demands of my family life limit the number of hours I’m able to work" "My family make it hard for me to do my job well" "I’m so tired from all the things I have to do at home that it’s hard to have the energy to do my job"
Items for chronic promotion focus (CPM) are "I frequently imagine how I will achieve my hopes and aspirations" "I often think about the person I would ideally like to be in the future""I typically focus on the success I hope to achieve in the future"
Items for chronic prevention focus (CPV) are "I often think about the person I am afraid I might become in the future" "I am anxious that I will fall short of my responsibilities and obligations" "In general, I am focused on preventing negative events in my life"
Items for challenge stress (CST) are "The number of projects and/or assignments I have""The amount of time I spend at work" "The volume of work that must be accomplished in the allotted time"
Items for hindrance stress (HST) are "The lack of job security I have" "The degree to which my career seems “stalled".
Reviewer 2 Report
The reviewer appreciates the research effort. There are some suggestions that would help the clarity and flow of the paper. Lines 25-27: "According to Gray, personality can be divided into anxiety tendency and impulse tendency on the basis of different nervous systems and functions of the brain, and Gray believes that different personality traits lead to diverse motivations and behaviors". Please move the references to this lines. In addition there are two sentences: "According to Gray, personality can be divided into anxiety tendency and impulse tendency on the basis of different nervous systems and functions of the brain. Gray(or He) believes that different personality traits lead to diverse motivations and behaviors". Also it is suggested to delete some words on line 29- "Based on previous theoretical studies, Higgins proposed the regulatory focus theory stressors" to "On previous theoretical studies, ...").
Line 41: "source of stress for individuals" should be "sources of stress". I am not sure what the authors mean by "organizational citizenship behavior and line 42 . Maybe needs some rephrasing.
Line 70: "sustainable employability the staff" should it be "sustainable employability of the staff"
Line 70: The expression :"When direct leaders adopt ethical leadership"- are the authors talking about first line supervisors and/or management?
Line 72: "alleviated and they will behave more actively in their work"- this is confussing - what does it mean to behave "more actively"?
Line 83: "lack of research outside of Western countries" should read "lack of research outside Western countries."
Lines 99-100: "Conflicts in work and family role demands relate to higher levels of stress.." should read "Conflicts between work and family role demands relate to higher levels of stress"
Line 108: "preserve resources through the substituting and transferring of resources" should read "preserve resources by substituting and transferring resources"
Lines 135 and 136- is there something missing at the end of line 135? Individuals adopt promotion or prevention self-regulation strategies relatively consistently that influence work attitudes such as organizational commitment" Is it "Individuals adopt promotion or prevention self-regulation strategies relatively that consistently influence work attitudes such as organizational commitment"
Line 157- "This section may be divided by subheadings" should be"This section is divided by subheadings (or sub-sections)".
Line 157- "It should provide a concise and precise" change to "It provides a concise and precise"
There are multiple examples, so please, review grammar.
Overall- there is no definition of specific "family roles". For study 1, please provide examples of the questionnaire/survey questions. They are discussed in later sections but the general description is missing from the study 1 methodology.
The data does not describe the specific role of the participant, particularly in the family (main provider?, single-no worries?). It seems that the study could have also expand to gender differences- are stressors perceived different by males and females in the same roles?
Also-line 382 talks about WIF but the wording is not clear.
Line 438- the words "should redesign" are too strong. Maybe use "should consider". In addition, some of the suggestions are already implemented in many countries (lines 439- 440)- e.g., on-line platform for virtual team members working from home- it is called telework.
Author Response
Point 1: 1)Lines 25-27: "According to Gray, personality can be divided into anxiety tendency and impulse tendency on the basis of different nervous systems and functions of the brain, and Gray believes that different personality traits lead to diverse motivations and behaviors". Please move the references to this lines. In addition there are two sentences: "According to Gray, personality can be divided into anxiety tendency and impulse tendency on the basis of different nervous systems and functions of the brain. Gray(or He) believes that different personality traits lead to diverse motivations and behaviors". Also it is suggested to delete some words on line 29- "Based on previous theoretical studies, Higgins proposed the regulatory focus theory stressors" to "On previous theoretical studies, ...").
2)Line 41: "source of stress for individuals" should be "sources of stress". I am not sure what the authors mean by "organizational citizenship behavior and line 42 . Maybe needs some rephrasing.
3)Line 70: "sustainable employability the staff" should it be "sustainable employability of the staff"
4)Line 70: The expression :"When direct leaders adopt ethical leadership"- are the authors talking about first line supervisors and/or management?
5)Line 72: "alleviated and they will behave more actively in their work"- this is confussing - what does it mean to behave "more actively"?
6)Line 83: "lack of research outside of Western countries" should read "lack of research outside Western countries."
7)Lines 99-100: "Conflicts in work and family role demands relate to higher levels of stress.." should read "Conflicts between work and family role demands relate to higher levels of stress"
8)Line 108: "preserve resources through the substituting and transferring of resources" should read "preserve resources by substituting and transferring resources"
9)Lines 135 and 136- is there something missing at the end of line 135? Individuals adopt promotion or prevention self-regulation strategies relatively consistently that influence work attitudes such as organizational commitment" Is it "Individuals adopt promotion or prevention self-regulation strategies relatively that consistently influence work attitudes such as organizational commitment"
10)Line 157- "This section may be divided by subheadings" should be"This section is divided by subheadings (or sub-sections)".
11)Line 157- "It should provide a concise and precise" change to "It provides a concise and precise"
12) Line 438- the words "should redesign" are too strong. Maybe use "should consider". In addition, some of the suggestions are already implemented in many countries (lines 439- 440)- e.g., on-line platform for virtual team members working from home- it is called telework.
Response 1: Grammar mistakes referred above have been corrected.
Line 26: "According to Gray, personality can be divided into anxiety tendency and impulse tendency on the basis of different nervous systems and functions of the brain. Gray believes that different personality traits lead to diverse motivations and behaviors".
Line 29:"On previous theoretical studies, Higgins proposed the regulatory focus theory stressors"
Line 41:"Work family conflict is one of the important source of stress."
Line 71:"Work pressure may threaten the sustainable employability of the staff"
Line 73:"When direct leaders adopt ethical leadership, pay attention to the interests of employees and provide support for leaders, employees' anxiety can be effectively alleviated and they will behave more better in their work"
Line 84:"Third, responding to the complaints of inconsistent cultural effects and lack of research outside Western countries"
Line 100: Conflicts between work and family role demands relate to higher levels of stress
Line 109:Resource Conservation Theory proposes that individuals tend to acquire and preserve resources by substituting and transferring resources.
Line 136:Individuals adopt promotion or prevention self-regulation strategies relatively that consistently influence work attitudes such as organizational commitment
Line 159:This section is divided by subheadings. It provides a concise and precise description of the experimental results, their interpretation as well as the experimental conclusions that can be drawn.
Line 455: "on-line platform for virtual team members working from home" has been replaced as "telework"
Point 2: For study 1, please provide examples of the questionnaire/survey questions.
Response 2: Measurement items and loading factors have been displayed by table 3 in line 282.
Items for Work interfering with family (WIF) are "When I get home from my job, I do not have the energy to do work around the house" "I spend so much time working that I am unable to get much done at home" "After work, I am often too tired to do things with my spouse or partner".
Items for family interfering with work (FIW) are "The demands of my family life limit the number of hours I’m able to work" "My family make it hard for me to do my job well" "I’m so tired from all the things I have to do at home that it’s hard to have the energy to do my job"
Items for chronic promotion focus (CPM) are "I frequently imagine how I will achieve my hopes and aspirations" "I often think about the person I would ideally like to be in the future"
"I typically focus on the success I hope to achieve in the future"
Items for chronic prevention focus (CPV) are "I often think about the person I am afraid I might become in the future" "I am anxious that I will fall short of my responsibilities and obligations" "In general, I am focused on preventing negative events in my life"
Items for challenge stress (CST) are "The number of projects and/or assignments I have"
"The amount of time I spend at work" "The volume of work that must be accomplished in the allotted time"
Items for hindrance stress (HST) are "The lack of job security I have" "The degree to which my career seems “stalled".
Reviewer 3 Report
It is an interesting topic. However, the overall manuscript is acceptable standards for publication in IJERPH. But before acceptance of this manuscript need some revisions. I explain some of my reservations with major revisions in detail below.
- Abstract: The abstract lack of clarity. In the abstract, the background/objective of study is not explained as well as the results and analysis and consolation is not explaining well in the abstract. I suggest you rewrite or improve the abstract of your study.
- Introduction: The introduction part of this study is too long and does not provide innovative information to support the rational of this study. It is suggested provides a more rational background of the gap of the study and research questions. As well as explain your research questions in detail that what research you are going to do? The below-mentioned papers will help you to write the introduction of your paper. Also suggest you to refer below mentioned papers into your paper.
- Literature Review: The literature review is missing in your manuscript. I suggest you to create heading with the name of literature review and express literature review in detail in this section.
- Theoretical Framework: Theoretical framework of this study is ok
- Hypothesis: The hypothesis part of this study is very weak. I suggest to the authors every hypothesis explain with the help of previous literature. Also suggest to the author integrate theoretical framework with hypothesis. Below mentioned paper will help you to write your hypothesis.
- Methodology: Research methodology of this study is not explained well. I explain some of my reservations in detail below.
- Which research approach authors used in this study?
- Which sampling techniques used for this study?
- Express in detail about the validity of the questions and scenario that authors used in this study?
- Analysis and Results: In this part there is no explanation of missing data how missing data was handled during the analysis of the data. It is suggested to you explain how do you handle missing data in this study.
- Discussion: This discussion section is ok
- Conclusion: The conclusion section is missing. It is suggested to the authors the conclusions must be explain in detail and interlinked with the discussion and literature review of this study. Moreover, the authors need to answer the key questions that you will raised in the revised manuscript. It is suggesting to the authors explain the novelty of the study. After these revisions this paper could be taken into consideration.
- Managerial implications: Please explain what level of managers as well as what kind of mangers (Functional managers) will apply your study at their profession and what will be the outcome of this implications.
- References: The reference style authors used in this study is not correct. It is suggested fallow the IJERPH Journal (MDPI publisher) reference style.
Author Response
Point 1:The reference style authors used in this study is not correct. It is suggested fallow the IJERPH Journal (MDPI publisher) reference style.
Response 1: Reference style has been corrected by downloading Endnote "MDPI style"
Point 2: Abstract: The abstract lack of clarity. In the abstract, the background /objective of study is not explained as well as the results and analysis and consolation is not explaining well in the abstract. I suggest you rewrite or improve the abstract of your study.
Response 2: Several sentences have been added to the abstract from line 18 to line 25 and line 28.
Reviewer 4 Report
The topic of work-family conflict is extensively covered in the scientific literature, however, this paper originally relates this concept to that of the Chronic regulatory focus.
The authors have prepared a theoretical framework not completely updated, they have not included and considered some significant research on the topic of WFC published more recently, starting from 2018, and easily identifiable in the databases.
The research design is overall well set up, although it would be appreciable that the authors justify the choice of the hospitality sector as a case study, especially with respect to the sectors investigated in the research already published, such as those they have referred to in the manuscript and carried out in the United States.
The methodological approach is clearly described and the results are well related to the empirical model.
However, in the conclusions it would be better to emphasize the innovative contribution of this research compared to other studies previously conducted in other socio-cultural and sectorial realities, specifying the collocation of this paper in relation to the state of the art.
For the reasons set out above, I advise authors to carry out a revision of the manuscript in order to further enhance its contents before its publication.
Author Response
Point 1:In the conclusions it would be better to emphasize the innovative contribution of this research compared to other studies previously conducted in other socio-cultural and sectorial realities, specifying the collocation of this paper in relation to the state of the art.
Response 1: Comparison to previous studies has been shown in line450 according to the suggestions on revision.
Round 2
Reviewer 1 Report
The paper looks good and thanks to authors for the hard work.
Please check the formatting of the whole paper. This is not like academic. For example, the limitation part doesn't need to mention heading and numbering. Moreover, include the future direction of the study.
Author Response
The number of limitation part has been deleted and according modification is added in line 470.
Practical Implications has been modified in line 453.
Reviewer 3 Report
This paper is on the quality standards of an academic paper. So, I accept this paper.
Author Response
Thank you for your acknowledgement.
Reviewer 4 Report
Compared to the first version of the manuscript, I still advise authors to update the literature review to include the most recent studies on this topic.
Author Response
The authors believe that the literature review has clearly explained the research background, significance and current research status.